# OLDIE BUT GOODIE: RE-ILLUMINATING LABEL PROPAGATION ON GRAPHS WITH PARTIAL FEATURES

## ABSTRACT

In real-world graphs, we often encounter missing feature situations where a few or the majority of node features, e.g., sensitive information, are missed. In such scenarios, directly utilizing Graph Neural Networks (GNNs) would yield sub-optimal results in downstream tasks such as node classification. Despite the emergence of a few GNN-based methods attempting to mitigate its missing situation, when only a few features are available, they rather perform worse than traditional structure-based models. To this end, we propose a novel framework that further illuminates the potential of classical Label Propagation (Oldie), taking advantage of Feature Propagation, especially when only a partial feature is available. Now called by Goodie[1], it takes a hybrid approach to obtain embeddings from the Label Propagation branch and Feature Propagation branch. To do so, we first design a GNN-based decoder that enables the Label Propagation branch to output hidden embeddings that align with those of the FP branch. Then, Goodie automatically captures the significance of structure and feature information thanks to the newly designed Structure-Feature Attention. Followed by a novel Pseudo-Label contrastive learning that differentiates the contribution of each positive pair within pseudo-labels originating from the LP branch, Goodie outputs the final prediction for the unlabeled nodes. Through extensive experiments, we demonstrate that our proposed model, Goodie, outperforms the existing state-of-the-art methods not only when only a few features are available but also in abundantly available situations.

## 1 INTRODUCTION

Graph embedding techniques have been favored since the early stage of the graph community and widely used until recently. Among various techniques, traditional graph embedding methods focus on how to leverage relational information in a given graph. To preserve its structural properties and information, they aim to obtain a computational and low-dimensional continuous vector for each node Perozzi et al. (2014a;b). Without the usage of features and solely resorting to structure information, obtained embeddings for each node would semantically contain their local neighborhood structure Cao et al. (2015); Ou et al. (2016).

At the same time, the recent success of Graph Neural Networks (GNNs) Kipf & Welling (2016a); Hamilton et al. (2017); Veličković et al. (2017) arises from their ability to jointly encode not only structure but also feature information into low-dimensional embedding space. GNNs first aggregate the messages, i.e., features from their neighbors, then update their representation iteratively Gilmer et al. (2017). Sharing the essence with the aforementioned graph embedding techniques, GNNs aim to solve downstream tasks such as node classification Kipf & Welling (2016a); Veličković et al. (2017), and link prediction Kipf & Welling (2016b); Zhang & Chen (2018) with its obtained embeddings.

However, one of the strong inductive biases GNNs hold is the observation of full features, which do not reflect real-world settings Taguchi et al. (2021); Jiang & Zhang (2020); Chen et al. (2020); Rossi et al. (2021). In real-world graphs, as features have characteristics of high dimensionality in most cases, the missing situation is easily observable. In practical applications, feature information

---

[1]Source code of Goodie can be found in the Supplementary Material.
[1]At low observed rates, GCNMF which assumes observation on each feature channel, could not output results due to the existence of a channel where all features are missed.

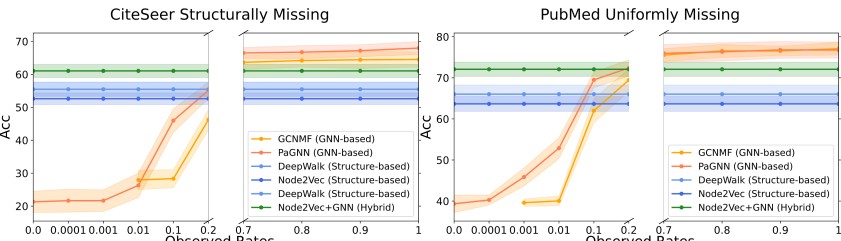

Figure 1: Node classification result of Structure-based, GNN-based, and Hybrid models on graphs with partial features. For graphs with severely missing features, structure-based models outperform[3].

may be missing to varying degrees, where it ranges from 0.01% - 99.99%. For example, in social networks, a small number of users, e.g., 20% tend to disclose their private and sensitive information such as income, and personal history Nulty (2008); Bollinger & Hirsch (2013); Emmanuel et al. (2021). Moreover, regarding the recent trend of incorporating graph structure in various domains, they severely suffer from missing feature scenarios. For the bio-medical domain, the missing rate can exceed 80% Yang et al. (2018); Zeisel et al. (2015); van Dijk et al. (2018); Qiu (2020) for scRNA-seq data. Also, in recommender system domains, where users and items are represented as nodes in bipartite graphs, the observed feature ratio can merely end up 1% to 5% Marlin et al. (2011); Ma et al. (2007) for the movie rating datasets.

In this regard, a few methods that first impute missing features and attempt to exploit GNN have been proposed Taguchi et al. (2021); Jiang & Zhang (2020); Chen et al. (2020). To cope with existing Graph Convolutional Network (GCN) Kipf & Welling (2016a), GCNMF Taguchi et al. (2021) pre-process feature imputation via assuming Gaussian Mixture Model, while PaGNN Jiang & Zhang (2020) adopts partial aggregation in the perspective of message propagation as like GCN. To verify their effectiveness, we empirically conducted experiments assuming real-world missing scenarios with traditional graph embedding techniques, DeepWalk Perozzi et al. (2014a), and Node2vec Perozzi et al. (2014b), including a hybrid model, Node2Vec+GNN. As shown in Figure 1, the interesting observations are as follows: **1)** When features are not or partially given, recent GNN-based methods deteriorate more than traditional graph embedding methods that solely use structure information. This is because when only a few features are available, heavily depending on this observed feature would hamper the generalization of the trained model. Thus, in such cases, rather utilizing structure-based models would be preferable. **2)** Hybrid model which initializes its feature via embeddings from Node2Vec outperforms traditional structure-based models. This tells us that implicitly utilizing features with equipping message-passing scheme is beneficial. **3)** However when features are provided more, GNN-based methods gain their potential to leverage feature information and eventually perform better than both structure-based and hybrid models. This provides us an insight that when features are abundant, explicitly utilizing them with a message-passing scheme is desirable.

We now conclude that different types of models are suited for different cases. However, in real-world scenarios, it is hard to decide which type of approach is appropriate in various missing situations. Another problem is the existing gap between traditional structure-based models and recent GNN-based models, especially when only a few features are available. Although the implicit hybrid model can mitigate the gap to some extent, it still has the pitfalls of incorporating explicit features. Thus, the following questions naturally emerge, how can we appropriately use the partial feature information while taking advantage of traditional structure-based graph embedding approaches? More generally, how can we adaptively use the feature information and combine it with the structure information in different cases?

Based on such motivation, we now propose a novel framework, called `Goodie` that takes an explicit hybrid approach to incorporate structure and feature information upon graphs with partial features. To do so, we especially utilize traditional Label Propagation (LP) Zhu (2005) and a very recently proposed algorithm, Feature Propagation (FP) Rossi et al. (2021) to cope with structure and feature information, respectively. As they propagate labels and features to their neighbors, we design an elegant way to combine both. More precisely, at the LP branch, we cope with a GNN-based decoder that transforms the LP's predicted logits into a hidden embedding vector. By doing so, the LP branch not only equips learnable parameters while training but also aligns with the hidden embedding space

originating from the FP branch. Then, we apply structure-feature attention to automatically obtain embeddings that have effectively "separated the wheat from the chaff" from each branch. After having obtained those embeddings, we apply a novel pseudo-label contrastive learning that utilizes the prediction originated from the LP branch, namely pseudo-labels. With extensive experiments, we demonstrate that `Goodie` outperforms a wide range of existing structure-based models and GNN-based models for both slightly-missing and severely-missing situations.

- We introduce a hybrid approach that bridges the gap between traditional structure-based models and recent GNN-based models, particularly effective when only limited features are available.
- We develop a Structure-Feature Attention module to effectively capture the importance of both structural and feature information in the model equipped with a novel Pseudo-Label Contrastive Learning which differentiates the contribution of the train and pseudo-label pairs.
- We demonstrate that our proposed model, `Goodie`, outperforms structure-based, hybrid, and GNN-based models on both slightly-missing ($0 \sim 10\%$) and severely-missing situations ($99.9 \sim 100\%$) in downstream tasks evaluated on various datasets.

## 2 RELATED WORK

**Label Propagation.** Given a graph with partial labels, Label Propagation (LP) Zhu (2005); Zhou et al. (2003); Wang & Zhang (2006) diffuses its known labels based on graph structure and predicts the labels of the unlabeled nodes. At the early stage of the graph community, thanks to its simplicity and intuitive approach to predicting labels, LP has been widely used for various tasks Zhang & Lee (2006); Raghavan et al. (2007). However, its popularity has declined as Graph Neural Networks are being introduced Kipf & Welling (2016a); Veličković et al. (2017); Hamilton et al. (2017) and recognized its power upon its ability to incorporate features and encode them with structure jointly. Recently, a few works have connected GNNs with LP Huang et al. (2020); Wang & Leskovec (2020); however, they fall short when features are not fully observed, e.g., real-world scenarios.

**Feature Propagation.** To mitigate missing feature problems in the graph domain, Feature Propagation (FP) Rossi et al. (2021) has been recently introduced. Namely, FP diffuses its known features to unknown features iteratively and reconstructs the feature matrix followed by GNN layers. From the perspective of minimizing Dirichlet energy, it naturally imputes nodes' missing features from their connected neighbors. By virtue of assortative environments on graphs Sen et al. (2008); Defferrard et al. (2016); Kipf & Welling (2016a); Lin & Cohen (2010), the imputed nodes eventually share similar features with their neighbors. Despite its promising performance and scalability, it still suffers from situations when only a few features are given, resulting in poor performance compared to simple LP.

**Other works handling Missing Features in Graphs.** As the missing feature situation is easily observed in real-world scenarios, in the graph domain, node features naturally contain missing elements. To handle such cases and relate to existing GNNs, several works have been proposed. GCNMF Taguchi et al. (2021) imputes missing features by assuming each feature channel follows the Gaussian distribution and suggests Gaussian Mixture Model that aligns with graph convolutional networks (GCN). Also, PaGNN Jiang & Zhang (2020) suggests a partial aggregation scheme from neighborhood reconstruction formulation. SAT Chen et al. (2020) jointly takes into account structure and attributes through distribution matching. Here, while SAT utilizes structure information as well as feature information, the main difference between our proposed model, `Goodie`, is the explicit usage of their embeddings and the control of their contribution. `Goodie` explicitly utilizes structure embeddings originated from LP, and controls its weight between feature embeddings obtained from FP. Although these methods showed their effectiveness in low missing rates, they can not cope with the situation when only a few features are available.

## 3 PROBLEM STATEMENT

**Graph.** Given a graph, $\mathcal{G} = (\mathcal{V}, \mathcal{E}, \mathbf{X})$, we denote $\mathcal{V} = \{v_1, ..., v_N\}$ as the set of nodes, $\mathcal{E} \subseteq \mathcal{V} \times \mathcal{V}$ set of edges, and $\mathbf{X} \in \mathbb{R}^{N \times F}$ feature matrix, where $F$ is the feature dimension on each node. $\mathbf{A} \in \mathbb{R}^{N \times N}$ denotes an adjacency matrix where $\mathbf{A}_{ij} = 1$ if $(v_i, v_j) \in \mathcal{E}$ and $\mathbf{A}_{ij} = 0$ otherwise. We denote $\mathcal{C}$ as the set of classes of nodes in $\mathcal{G}$.

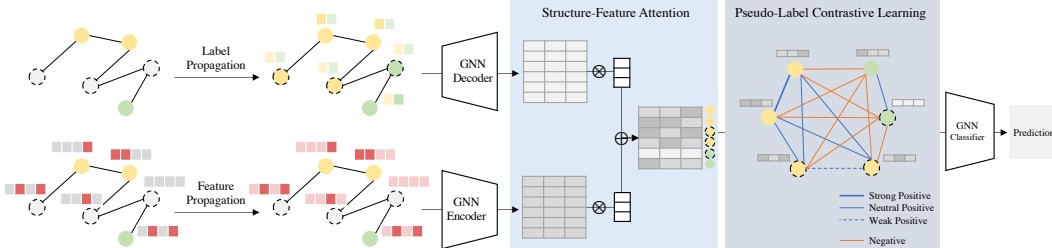

Figure 2: The overall architecture of `Goodie`. Given a graph with partially observed features in semi-supervised settings, we first obtain embeddings from LP and FP branches. Followed by Structure-Feature Attention, we obtain embeddings that contain the significance of each branch. With pseudo-labels originating from the LP branch, we further improve the embeddings via pseudo-label contrastive learning and make the final prediction.

**Task: Node Classification and Link Prediction with Partially Observed Features.** Given a graph $\mathcal{G}$ with $X$ containing missing elements, we aim to learn a GNN-based decoder, encoder, and classifier that works well on node classification and link prediction.

## 4 PROPOSED FRAMEWORK: GOODIE

In this section, we present a novel framework that further illuminates the potential of LP with FP on graphs with partially observed features. We first demonstrate how each LP and FP branch output embeddings of each node (Sec 4.1), and how the embeddings further capture the significance of each branch via the structure-feature attention module (Sec 4.2). Then, we introduce a novel pseudo-label contrastive learning that further improves the quality of embeddings with its scaled version (Sec 4.3), followed by the overall model training process (Sec 4.4). Figure 2 shows the overall architecture of `Goodie`.

### 4.1 LP & FP BRANCH

To leverage given labels in semi-supervised settings, we begin with the LP branch. Here, to align with symmetric normalized GNNs Kipf & Welling (2016a), we utilize the transition (affinity) matrix as a symmetric normalized adjacency matrix, $\hat{\mathbf{A}}_{sym} = \tilde{\mathbf{D}}^{-1/2}\tilde{\mathbf{A}}\tilde{\mathbf{D}}^{-1/2}$, where $\tilde{\mathbf{D}} = \sum_i \tilde{\mathbf{A}}_i$ is the degree matrix, and $\tilde{\mathbf{A}} = \mathbf{A} + \mathbf{I}$ is the adjacency matrix with self-loops. Also, with the initial label matrix, $\mathbf{Y}^{(0)} = [\mathbf{y}_1^{(0)}, \cdots, \mathbf{y}_m^{(0)}, \cdots, \mathbf{y}_N^{(0)}]$ consisting one-hot vectors for the labeled nodes indexed $1, \cdots, m$ and zero vectors for the remaining unlabeled nodes, Label Propagation Zhu (2005) is formulated in two steps as follows:

$$\begin{aligned} \mathbf{Y}^{(k+1)} &= \hat{\mathbf{A}}_{sym}\mathbf{Y}^{(k)}, \\ \mathbf{y}_i^{(k+1)} &= \mathbf{y}_i^{(0)}, \forall i \leq m. \end{aligned} \tag{1}$$

where $\mathbf{Y}^{(k)}$ denotes label matrix at iteration $k$. Intuitively, LP first propagates its known labels to its neighbors and replaces the propagated labels with initial labels for labeled nodes. Owing to its convergence property Zhu (2005); Zhou et al. (2003), after $K$ iterations, we obtain its converged label matrix, $\mathbf{Y}^{(K)}$, i.e., logits for each node.

However, naively concluding $\mathbf{Y}^{(K)}$ as the final prediction possess two main weakness: 1) it does not take feature information into account; 2) it lacks trainable parameters that are well-suited for downstream tasks. To alleviate such weaknesses, we design a GNN-based decoder that enables the converged label matrix to be represented in embedding space which later interacts with feature embeddings. Formally, the embeddings from the LP branch can be expressed as:

$$\mathbf{H}^{\text{LP}} = \sigma(\hat{\mathbf{A}}_{sym}\hat{\mathbf{Y}}\mathbf{W}_{\text{LP}}), \quad \hat{\mathbf{Y}} = \mathbf{Y}^{(K)} \tag{2}$$

where $\mathbf{H}^{\text{LP}} \in \mathbb{R}^{N \times D}$ denotes the node embedding matrix with hidden dimension, $D$, $\mathbf{W}_{\text{LP}} \in \mathbb{R}^{|\mathcal{C}| \times D}$ is the weight matrix that transforms logits into embedding space, and $\sigma$ is nonlinear activation function, ReLU. Here, among various GNNs, we adopt a message-passing scheme of GCN Kipf & Welling (2016a) as the backbone throughout the paper.

In the FP branch, considering we are dealing with a missing feature situation, we utilize the recent simple and efficient feature imputation method, Feature Propagation Rossi et al. (2021). Sharing the essence of LP but from a feature perspective, FP can be formulated as follows:

$$\mathbf{X}^{(k+1)} = \hat{\mathbf{A}}_{sym} \mathbf{X}^{(k)},$$
$$\mathbf{x}_{i,d}^{(k+1)} = \mathbf{x}_{i,d}^{(0)}, \forall i \in \mathcal{V}_{known,d}, \forall d. \tag{3}$$

where $\mathbf{X}^{(k)}$ denotes feature matrix at iteration $k$, and $\mathcal{V}_{known,d}$ is a set of nodes on which $d$-th channel feature values are *known*. Similar to LP, FP first propagates its known features to its neighbors and replaces the propagated features with initial features for the nodes with known features.

With its convergence property Rossi et al. (2021), after $K$ iterations, we obtain its converged feature-imputed matrix, $\mathbf{X}^{(K)}$, i.e., feature vectors for each node. Now, to obtain embeddings from the FP branch, it can be expressed as:

$$\mathbf{H}^{\text{FP}} = \sigma(\hat{\mathbf{A}}_{sym} \hat{\mathbf{X}} \mathbf{W}_{\text{FP}}), \quad \hat{\mathbf{X}} = \mathbf{X}^{(K)} \tag{4}$$

where $\mathbf{H}^{\text{FP}} \in \mathbb{R}^{N \times D}$ denotes the node embedding matrix, and $\mathbf{W}_{\text{FP}} \in \mathbb{R}^{F \times D}$ is the weight matrix that transforms features into low-dimensional embedding space.

## 4.2 STRUCTURE-FEATURE ATTENTION

It is important to note that we are facing a missing feature situation, which sometimes has low and sometimes high missing rates. In this regard, we want our model to naturally capture the significance and reflect the structure and feature information without any manual intervention. Thus, with the two embedding matrices, one containing structural information and the other containing feature information, we now proceed to apply Structure-Feature Attention. The core idea of Structure-Feature Attention is to automatically capture where the node should contain more of the structure information or the feature information. More precisely, as depicted in Figure 3, if features are not or partially available, embedding from the LP branch has to be more reflected. On the other hand, if features are abundant, embedding from the FP branch has to be more reflected.

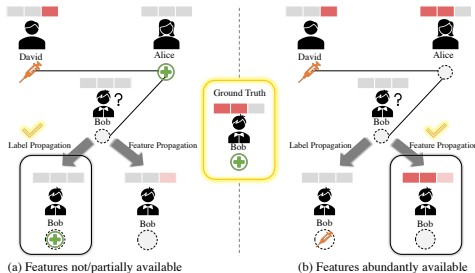

Figure 3: Real-world patient state prediction when either LP or FP information is significant. The significance of each branch would vary depending on its feature availability.

Formally, with attention coefficients $\alpha_{i,\text{LP}}, \alpha_{i,\text{FP}}, \forall i$ we aim to obtain node $i$'s embeddings as follows:

$$\mathbf{z}_i = \alpha_{i,\text{LP}} \mathbf{h}_i^{\text{LP}} + \alpha_{i,\text{FP}} \mathbf{h}_i^{\text{FP}} \tag{5}$$

where $\mathbf{z}_i \in \mathbb{R}^D$ denotes the embedding of node $i$ that now contains the significance of each structure and feature information. Here, attention coefficients can be obtained as follows:

$$\alpha_{i,LP} = \frac{\exp(\text{LeakyReLU}(\mathbf{a}^\top \mathbf{h}_i^{\text{LP}}))}{\exp(\text{LeakyReLU}(\mathbf{a}^\top \mathbf{h}_i^{\text{LP}})) + \exp(\text{LeakyReLU}(\mathbf{a}^\top \mathbf{h}_i^{\text{FP}}))}$$
$$\alpha_{i,FP} = \frac{\exp(\text{LeakyReLU}(\mathbf{a}^\top \mathbf{h}_i^{\text{FP}}))}{\exp(\text{LeakyReLU}(\mathbf{a}^\top \mathbf{h}_i^{\text{LP}})) + \exp(\text{LeakyReLU}(\mathbf{a}^\top \mathbf{h}_i^{\text{FP}}))} \tag{6}$$

where $\mathbf{a} \in \mathbb{R}^D$ is parameterized attention vector, and LeakyReLU denotes nonlinear activation function (with negative slope $\alpha = 0.3$).

Now, we compute cross-entropy loss with given train labels with GNN-based classifier as:

$$\mathcal{L}_{\text{ce}} = \sum_{v \in \mathcal{V}_{tr}} \sum_{c \in \mathcal{C}} CE(\mathbf{p}_v[c]), \mathbf{P} = \text{softmax}(\sigma(\hat{\mathbf{A}}_{sym}\mathbf{Z}\mathbf{W}_{\text{cls}})) \tag{7}$$

where $\mathcal{V}_{tr}$ denotes a set of train nodes, $\mathbf{P}$ is predicted class probability matrix, $\mathbf{W}_{\text{cls}} \in \mathbb{R}^{D \times |\mathcal{C}|}$ is the weight matrix that transforms low-dimensional vector into class-dimensional space, and $CE(\cdot)$ denotes cross-entropy loss.

### 4.3 PSUEDO-LABEL CONTRASTIVE LEARNING

Next, with embedding that contains the significance of the LP branch (structural information) and FP branch (feature information), we further leverage the potential of the LP branch. In other words, by exploiting additional supervision from LP, Goodie can enhance the learning performance under limited supervised label information, i.e., semi-supervised settings. More precisely, thanks to *pseudo*-labels, which are the

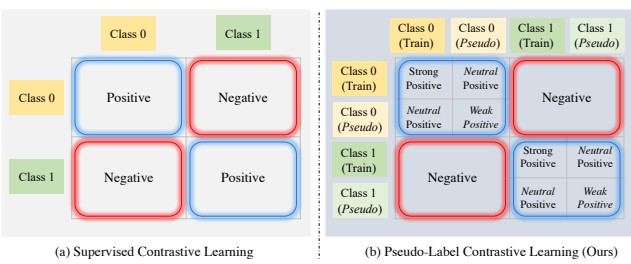

Figure 4: Comparison with Supervised Contrastive Leaning and Pseudo-Label Contrastive Learning.

predicted labels for unlabeled nodes based on the maximum index of the prediction logits for each node, the objective is to refine the embeddings and ensure that the embeddings for nodes with the same label are similar while those with different labels are dissimilar.

Motivated by recent contrastive learning in the computer vision domain that equips supervised signals, we aim to further extend SupCon Khosla et al. (2020) loss which is expressed as:

$$\mathcal{L}_{\text{sup}} = \sum_{i \in I} \frac{-1}{|P(i)|} \sum_{p \in P(i)} \log \frac{\exp(\mathbf{z}_i \cdot \mathbf{z}_p / \tau)}{\sum_{a \in A(i)} \exp(\mathbf{z}_i \cdot \mathbf{z}_a / \tau)} \tag{8}$$

where $\mathbf{z}_i$ denotes embedding of index $i$, $P(i) \equiv \{p \in A(i) : \mathbf{y}_p = \mathbf{y}_i\}$ is the set of all positive indices with its cardinality $|P(i)|$, $A(i) \equiv \{1, \cdots, N\}^4 \backslash \{i\}$, and $\tau \in \mathbb{R}^+$ is a temperature hyperparameter.

However, directly utilizing SupCon loss with incorporating *pseudo*-labels would face a main challenge. That is, considering we included *pseudo*-labels that possess uncertainty compared to given train labels, the significance of each positive pair would have to be different.

Now, to overcome the above challenge, we incorporate a weight parameter, $w_{ip}$ that generalizes the SupCon loss in the perspective of semi-supervised settings that utilize pseudo-labels. Formally, we define a pseudo-label contrastive loss, called PseudoCon as:

$$\mathcal{L}_{\text{pseudo}} = \sum_{i \in I} \frac{-1}{|P(i)|} \sum_{p \in P(i)} w_{ip} \cdot \log \frac{\exp(\mathbf{z}_i \cdot \mathbf{z}_p / \tau)}{\sum_{a \in A(i)} \exp(\mathbf{z}_i \cdot \mathbf{z}_a / \tau)} \tag{9}$$

Here, the weight parameter $w_{ip}$ is designed to achieve the following goals illustrated in Figure 4: (1) **Strong Positive**: if the node pairs consist of given train labels only, it should seamlessly align with supervised contrastive learning, (2) **Neutral Positive**: if the node pairs are consisting one given train label and one pseudo-label, it should reflect its uncertainty, and naturally, the weights should be less than (1). Lastly, (3) **Weak Positive**: if the node pairs consist of the pseudo-labels only, which contain more uncertainty, the weights should be the least among the above cases. Based on these goals, we define the weight parameter, $w_{ip}$ as:

---

[4]Here, as we do not require multiview originated from augmentation, in this paper, we consider single-view for each node.

$$w_{ip} = \begin{cases} 1, & \text{if } i, p \in \hat{Y}_{train} \\ \tilde{y}_p, & \text{if } i \in \hat{Y}_{train} \text{ and } p \in \hat{Y}_{pseudo} \\ \tilde{y}_i, & \text{if } i \in \hat{Y}_{pseudo} \text{ and } p \in \hat{Y}_{train} \\ \tilde{y}_i * \tilde{y}_p, & \text{if } i, p \in \hat{Y}_{pseudo} \end{cases} \tag{10}$$

where $\hat{Y}_{train}, \hat{Y}_{psuedo}$ denotes a set of indices given for train nodes and indices for the rest of nodes, respectively, and $\tilde{y}_\ell = max(\text{softmax}(\frac{\hat{\mathbf{Y}}_\ell}{\tau}))$ is the prediction probability of node $\ell \in 1, \cdots, N$, derived from the logits from LP branch, in Equation (2). Intuitively, this design further generalizes SupCon Khosla et al. (2020) loss with the usage of *pseudo*-labels and enables us to meet goals (1), (2), and (3). Also, by the usage of prediction probability and by virtue of its values' range, i.e., $0 \leq \cdot \leq 1$, the following inequality naturally holds that aligns with our motivation:

$$0 \leq \underbrace{\tilde{y}_i * \tilde{y}_p}_{\textbf{Weak Positive}} < \underbrace{\tilde{y}_i, \tilde{y}_p}_{\textbf{Neutral Positive}} \leq \underbrace{1}_{\textbf{Strong Positive}} \tag{11}$$

**SCALABILITY.** One challenge that occurs from introducing pseudo-label contrastive learning would be calculating the node embedding similarity matrix in Equation (9). In other words, from the perspective of full-batch training, the dot product between every node pair would cost $\mathcal{O}(N^2)$, which is not desirable in real-world large graphs. Thus, to cope with a large graph, we additionally propose a scalable version in pseudo-label contrastive learning that utilizes class prototypes. The motivation for using class prototypes is to abbreviate embeddings of nodes that share a common property, i.e., class information into one representative embedding. Here, solely incorporating nodes with given train labels would not sufficiently contain class information due to their limited number of samples. Again, thanks to pseudo-labels that provide additional supervision, we utilize pseudo-label information while generating class prototypes as follows:

$$\mathbf{z}^c = \frac{1}{|\hat{Y}^c|}\Big( \underbrace{\sum_{i \in \hat{Y}^c_{train}} 1 \cdot \mathbf{z}_i}_{\textbf{Strong}} + \underbrace{\sum_{j \in \hat{Y}^c_{pseudo}} \tilde{y}_j \cdot \mathbf{z}_j}_{\textbf{Neutral \& Weak}} \Big) \tag{12}$$

where $\mathbf{z}^c \in \mathbb{R}^D$ denotes the prototype embedding for class $c$, $\hat{Y}_c \equiv \{\hat{Y}^c_{train} \cup \hat{Y}^c_{pseudo}\}$ is set of node indices that belong to class $c$, and $\mathbf{z}_i$ is the embedding of node $i$ obtained from Equation (5). It is important to note that class prototype embedding is generated in a way that naturally controls pseudo-label uncertainty. In other words, as in the viewpoint of weighted sum, even if nodes belong to the same class, node embeddings from pseudo-labels (**Neutral & Weak**) are less reflected than those from train labels (**Strong**). With such class prototypes, we now calculate the scaled version of pseudo-label contrastive loss as follows:

$$\mathcal{L}_{\text{pseudo}} = -\sum_{c \in C} \log \frac{1}{\sum_{b \in B(c)} \exp(\mathbf{z}^c \cdot \mathbf{z}^b / \tau)} \tag{13}$$

where $B(c) \equiv \{1, \cdots, |\mathcal{C}|\}\setminus\{c\}$. Now, compared to the original pseudo-label contrastive loss in Equation (9), the computational cost of calculating the similarity matrix has been reduced from $\mathcal{O}(N^2)$ to $\mathcal{O}(|\mathcal{C}|^2)$ thanks to the use of class prototypes. By incorporating this scaled version loss, we expect each class prototype to be distinct from the others, i.e., implicitly pushing apart nodes from different classes.

### 4.4 MODEL TRAINING

To sum up, the overall training process of `Goodie` consists of two loss functions, $\mathcal{L}_{ce}$ (in Equation (7)), and $\mathcal{L}_{pseudo}$ (in Equation (9), 13) and finally represented as:

$$\mathcal{L}_{\text{final}} = \mathcal{L}_{\text{ce}} + \lambda \mathcal{L}_{\text{pseudo}} \tag{14}$$

where $\lambda$ is the hyperparameter that controls the contribution of pseudo-label contrastive loss. In the experiments, we utilized the scaled version of $\mathcal{L}_{\text{pseudo}}$ in large graph datasets such as Coauthor Physics and OGBN-Arxiv.

# 5 EXPERIMENTS

**DATASETS.** We evaluate our proposed framework on seven benchmark datasets: Cora, CiteSeer, PubMed Lin & Cohen (2010), WikiCS Mernyei & Cangea (2020), Coauthor CS, Coauthor Physics Sinha et al. (2015), and OGBN-Arxiv Hu et al. (2020). The statistics of each dataset can be found in Table 1. For more details, refer to Appendix A.3.

Table 1: Statistics of datasets.

| Dataset | #Nodes | #Edges | #Features | #Classes |
|---|---|---|---|---|
| Cora | 2,485 | 5,069 | 1,433 | 7 |
| CiteSeer | 2,120 | 3,679 | 3,703 | 6 |
| PubMed | 19,717 | 44,324 | 500 | 3 |
| WikiCS | 11,701 | 148,555 | 300 | 10 |
| Coauthor CS | 18,333 | 81,894 | 6,805 | 15 |
| Coauthor Physics | 34,493 | 247,962 | 8,415 | 5 |
| OGBN-Arxiv | 169,343 | 1,166,243 | 128 | 40 |

**EXPERIMENTAL SETTINGS AND EVALUATION METRICS.** Extending FP's setting Rossi et al. (2021), we evaluated our model with the above models in various missing rates, $mr$ ranging from low $mr$, $0\%, 10\%, \cdots$, to high $mr$, $99.99\%$ and even $100\%$. To cope with real-world missing scenarios, following Taguchi et al. (2021); Rossi et al. (2021), we masked the features in two ways. First, in a 1) *uniformly missing scenario*, among feature matrices, we randomly select elements with the ratio of $mr$, then mask the selected elements with unknown values (zeros). Second, in a 2) *structurally missing scenario*, we randomly select nodes with the ratio of $mr$, then mask the whole features of selected nodes with unknown values (zeros). For the node classification splits, we followed Rossi et al. (2021), which assigns 20 nodes per each class in the training set, a total of 1,500 nodes in the validation set, and the remaining nodes to the test set, except OGBN-Arxiv which has the fixed splits. For the evaluation metrics, we use accuracy (Acc) for the node classification and report the mean and standard deviations for 10 different random seeds. For the link prediction splits, we followed GCNMF's setting Taguchi et al. (2021) that randomly chose 10%, 5% edges for testing and validation, and the rest 85% for training. We utilized Graph Auto-Encoder (GAE) Kipf & Welling (2016b) for the base model and used hidden dimensions as 32, and 16 for the first and second layers of GCN.

## 5.1 PERFORMANCE ANALYSIS

**Node Classification.** Figure 8 shows the overall performance of `Goodie` with baseline models in various missing scenarios ranging from when features are not available (i.e., $mr = 1.0$) to when full features are available (i.e., $mr = 0.0$). We have the following observations: **1)** `Goodie` generally performs well not only on partial features but also on abundant features. More precisely, while recent GNN-based models, e.g., GCNMF, PaGNN, and FP fail severely due to limited information of features, `Goodie` outperforms those GNN-based models in the

Table 2: Performance (%) of `Goodie` on missing rate 0%, 50%, 99.99%, 100% with their relative drop on 0% (Full Features).

| Dataset | Full Features | 50 % Missing | 99.99% Missing | 100% Missing |
|---|---|---|---|---|
| Cora | 81.23 | 80.09(-1.14%) | 79.58(-1.65%) | 79.15(-2.08%) |
| CiteSeer | 68.42 | 67.1(-1.32%) | 66.44(-1.98%) | 66.11(-2.31%) |
| PubMed | 76.16 | 75.93(-0.23%) | 75.54(-0.62%) | 75.48(-0.68%) |
| WikiCS | 74.74 | 73.93(-0.81%) | 70.28(-4.46%) | 69.95(-4.79%) |
| Co. CS | 89.11 | 88.54(-0.57%) | 76.38(-12.73%) | 78.77(-10.34%) |
| Co. Physics | 92.24 | 92.12(-0.12%) | 87.86(-4.38%) | 88.02(-4.22%) |
| OGBN-Arxiv | 70.06 | 69.5(-0.56%) | 76.38(-1.02%) | 76.38(-0.92%) |

Table 3: Performance (%) of `Goodie` on node classification task with other baselines at $mr = 0.9999$. (OOM: Out of Memory on 24GB RTX 3090.)

| Dataset | PaGNN | GCN-LPA | C&S | LP | FP | Goodie |
|---|---|---|---|---|---|---|
| Cora | 25.1±4.61 | 31.88±2.17 | 60.02±5.06 | 74.77±1.00 | 55.89±7.46 | **79.58**±1.01 |
| CiteSeer | 22.6±1.77 | 24.24±1.07 | 65.55±1.61 | 66.15±1.67 | 51.06±4.17 | **66.44**±1.41 |
| PubMed | 40.26±1.22 | 41.17±1.18 | 58.98±10.05 | 72.32±4.35 | 73.18±1.51 | **75.54**±0.65 |
| WikiCS | 59.87±2.06 | 50.23±4.61 | 53.99±8.16 | 62.39±3.03 | 64.36±9.22 | **70.28**±2.57 |
| Co. CS | 27.79±2.87 | 36.58±1.53 | 53.99±8.16 | 76.54±1.52 | **78.54**±0.63 | 76.38±1.65 |
| Co. Physics | 44.16±5.89 | 53.34±1.44 | 45.03±23.68 | 85.86±1.91 | **87.92**±1.55 | 87.86±1.61 |
| OGBN-Arxiv | 44.69±0.41 | 22.15±5.86 | 67.72±0.14 | 67.36±0.0 | 63.72±0.54 | **69.04**±0.14 |
| Average | 38.78 | 37.08 | 57.90 | 72.20 | 67.81 | **75.02** |

early stage of observed rates. A further appeal of `Goodie` is its ability to cope with more of the features when they are getting abundant, which enables it to perform well on low missing scenarios as well. **2)** Such tendency can be specified in Table 2, where only a small percentage, e.g., $0.7 \sim 4.8\%$ drop is occurred in 100% missing scenarios. Here, it is worth noting that even though the drop rate is relatively high in the Coauthor CS dataset, performance at 100% Missing is still higher than Label Propagation, which tells us `Goodie` performs LP as a lower bound. Table 3 again shows our model's robustness on harsh missing cases compared to other baselines. **2)** We observe both the strength and weaknesses of traditional and GNN-based methods. The former, e.g., LP, Node2Vec lacks the ability to make use of features, and the latter, e.g., GCN, GCNMF, and FP especially become less powerful when features are not given ideally. **3)** Among hybrid models, the model that

utilized Label Propagation in their design, e.g., C & S survives better than recent GNN-based models, especially in partial feature scenarios. This is because they explicitly utilize structure information during the post-processing step, which is crucial when features are not given ideally. However, its performance is rather limited due to the absence of a module that imputes features and the sequential mechanism that depends on prior information. **4)** Also, we note that Node2Vec+GNN performs better than solely using Node2Vec, which tells us the effectiveness and importance of aggregating neighbor's information. **5)** As we can observe in Figure 8, the point when GNN-based models start to take effect varies upon missing situations and every dataset. This brings us to the necessity of automatic design, i.e., without manual intervention, that naturally captures the significance of each structure and feature information. For link prediction results, please refer to Appendix A.5

## 5.2 ABLATION STUDIES

### 5.2.1 EFFECTIVENESS OF STRUCTURE-FEATURE ATTENTION.

In this section, we first verify whether attention design is appropriate for current missing situations. Figure 5 shows the node classification results in high and low missing rates with its variants. **Random** denotes the random half of the nodes take embeddings from the LP branch, and the rest half takes the embeddings of the FP branch, where two pieces of information are utilized without any considerations. **Sum** and **Mean** are the sum and mean operation of two embeddings, respectively, enabling a node to take into account both structure and feature information. **Concat** denotes the concatenation of two embeddings from each branch followed by a GNN-based classifier.[5] We have the following observations: **1)** **Random** does not perform well compared to other cases due to its randomness in incorporating merely one of the structure and feature embeddings. **2)** Despite its simple and intuitive way of reflecting both information, **Sum** and **Mean** results in sub-optimal performance, e.g., the observed rate of 0.0001 in PubMed Strucutrally Missing situation. **3)** Now, coping with trainable parameters, **Concat** performs better than aforementioned operations, but again falls short in certain observed rates. We conjecture that its implicit way of reflecting still remains challenging to minimize the impact of harmful information. e.g., more feature information when only a few features are available. **4)** As **Attention** *(Ours)* operation generally performs well on various missing scenarios, we now further verify whether the **Attention** *(Ours)* module is capturing each of the branch's information appropriately. In Figure 6, we observe that score of attention on a feature is getting higher as more features are involved. This coincides with our model design motivation to automatically capture the significance of structure or feature information in diverse missing scenarios. For the effectiveness of Pseudo-label contrastive learning please refer to Appendix B.0.3

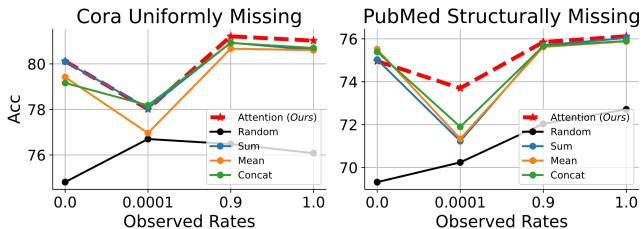

Figure 5: Effectiveness of Structure-Feature Attention with its replacements. Accuracy at each observed rate is the mean value of 10 different random seeds.

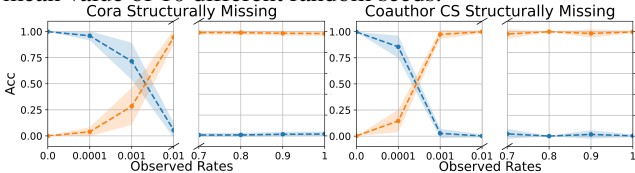

Figure 6: Attention score of coefficients, $\alpha_{LP}$, $\alpha_{FP}$ which is responsible for capturing structure and feature information, respectively. The value denotes the mean of each attention channel on total nodes with its standard deviation.

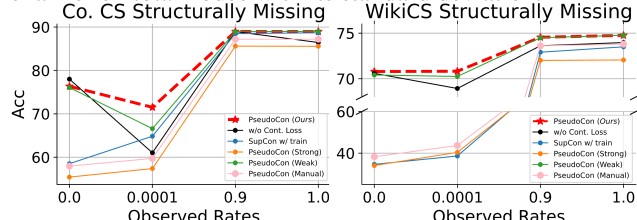

Figure 7: Effectiveness of Pseudo-Label Contrastive Learning with its replacements. Accuracy at each observed rate is the mean value of 10 different random seeds.

---

[5]Here, to minimize the effect of Pseudo-label contrastive loss, we set loss controlling parameter, $\lambda$ as 0 to solely focus on the current module.

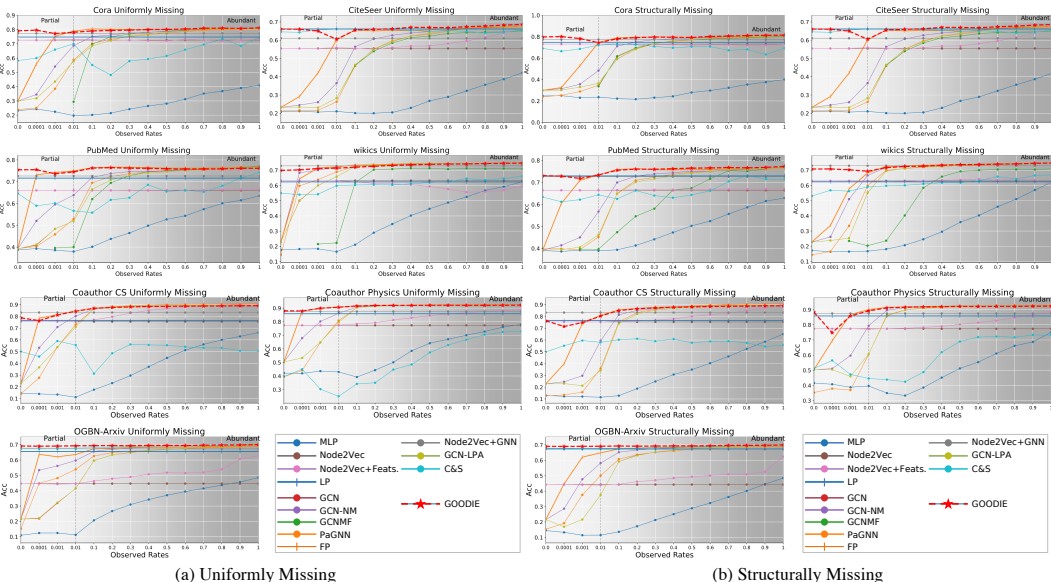

(a) Uniformly Missing          (b) Structurally Missing

Figure 8: Overall node classification accuracy of *Uniformly* and *Structurally* missing scenarios. The background color gets darker when features are abundant. All experiments in each observed rate are repeated 10 times with different random seeds.

## 5.3 SENSITIVITY ANALYSIS

For sensitivity analysis, please kindly refer to Appendix B

## 6 CONCLUSION

In this paper, we proposed a novel hybrid approach to bridge the gap between the traditional structure-based model and the recent GNN-based model, especially in graphs with partial features. Based on the empirical observation that recent GNN-based imputation models deteriorate more than traditional graph embedding methods, we proposed `Goodie` that leverages the hidden potential of the classic structure-based model, Label Propagation. `Goodie` first obtain embeddings from the LP branch thanks to a simple GNN-based decoder that enables alignment with those of the FP branch. Passed through the Structure-Feature Attention module, we naturally obtain embeddings that *separated the wheat from the chaff* contain appropriate structure and feature information. We further propose Pseudo-Label Contrastive Learning that leverages the predicted probability originating from the LP branch which differentiates each positive pair with train and pseudo labels. Through extensive experiments regarding various missing scenarios, we empirically show the robust performance on not only slightly-missing but also severely-missing situations.

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

## A  Supplement for experiments

### A.1  Compared Methods

Considering we are facing a missing feature situation where features are partially observed in semi-supervised settings, we compare `Goodie` with the following models that are widely used in graphs with missing scenarios:

(1) Feature or Structure-based models

- **MLP**: It is simple 2-layer Multi-Layer Perceptron with ReLU. For the missing features, we imputed them with zeros.

- **Node2Vec** Perozzi et al. (2014b): It learns node embedding via random walks with skip-gram. Since it does not consider feature information, it is not sensitive to missing rates. We also experimented **Node2Vec+Feats.**, which leverages given partial features.

- **Label Propagation** Zhu (2005): It directly propagates the given labels to their neighbors until they converge. As LP only relies on structure information, it is not sensitive to missing rates.

(2) GNN-based models

- **GCN** Kipf & Welling (2016a): It is simple 2-layer Graph Convolutional Network. For the missing features, we imputed them with zeros which naturally performs well as mentioned in Rossi et al. (2021).

- **GCN-NM** Rossi et al. (2021): This is a graph-based imputation method that imputes missing features with the mean of its neighbors. This corresponds to the first-order approximation of FP.

- **GCNMF** Taguchi et al. (2021): It is an end-to-end GNN-based model that imputes missing features by assuming Gaussian Mixture Model that aligns with GCN.

- **PaGNN** Jiang & Zhang (2020): It performs a partial message-passing scheme via propagating observed features only.

- **FP** Rossi et al. (2021): It is a state-of-the-art model that propagates given features through neighbors and replaces observed ones with their original ones in terms of minimizing the Dirichlet energy.

(3) Hybrid models

- **Node2Vec+GNN**: It regards the node embeddings from Node2Vec as features of GNN. It then follows the message-passing scheme of GCN.

- **GCN-LPA** Wang & Leskovec (2020): It connects GCN with Label Propagation via introducing trainable edges and trains them jointly.

- **Correct and Smooth (C&S)** Huang et al. (2020): It connects simple MLP with Label Propagation by using MLP to obtain base prediction and utilizes LP as a post-processing step. We used the Linear model as a base predictor.

### A.2  Implementation Details.

While training, we use the Adam Kingma & Ba (2014) optimizer where the learning rate is 0.005 with a hidden dimension set as 64, and the dropout rate set as 0.5 across all the compared methods. With a maximum of 10,000 epochs, we used early stopping with patience 200 and evaluated their best model based on their validation accuracy. For Node2Vec, we used the return and in-out parameters both 1.0 and for Label Propagation, we selected $\alpha$ in $\{0.8, 0.9, 0.99, 0.999\}$ which is also used in our model, `Goodie`. For GNN-based methods and hybrid methods, e.g., GCNMF, PaGNN, GCN-LPA, and C&S we adopt their hyperparameter settings reported in their papers. For the number of iterations, $K$, we used 50 for Label Propagation and 40 for Feature Propagation, which was enough for labels and features to be converged as mentioned in Rossi et al. (2021). For `Goodie`, we searched loss controlling parameter $\lambda$ in $\{0.00001, 0.0001, \cdots, 1, 10\}$ with temperature $\tau$ in psuedo-label contrastive learning set as 0.01. (Refer to Appendix A.4 for detailed setting.)

## A.3 DATASETS AND CODES

For experiments, we used seven graph benchmark datasets, ranging from Citation network (Cora, CiteSeer, PubMed), Coauthor network (CS, Physics), Wikipedia network (WikiCS), and Citation network from Open Graph Benchmark (OGBN-Arxiv). URL link to each dataset can be found in Table 4. For the baseline models, Table 5 shows the URL link to its implementation. Among baselines, since PaGNN Jiang & Zhang (2020) did not release public code, we referred to FP Rossi et al. (2021) authors' implementation.

Table 4: URL link to each dataset.

| Dataset | URL link to the dataset |
|---|---|
| Cora | https://github.com/shchur/gnn-benchmark |
| CiteSeer | https://github.com/shchur/gnn-benchmark |
| PubMed | https://github.com/shchur/gnn-benchmark |
| WikiCS | https://github.com/pmernyei/wiki-cs-dataset |
| Coauthor CS | https://github.com/shchur/gnn-benchmark |
| Coauthor Physics | https://github.com/shchur/gnn-benchmark |
| OGBN-Arxiv | https://ogb.stanford.edu/docs/nodeprop/#ogbn-arxiv |

Table 5: URL link to each model's code.

| Dataset | URL link to the code |
|---|---|
| Node2Vec | https://github.com/aditya-grover/node2vec |
| FP | https://github.com/twitter-research/feature-propagation |
| GCNMF | https://github.com/marblet/GCNmf |
| GCN | https://github.com/tkipf/pygcn |
| GCN-NM | https://github.com/twitter-research/feature-propagation |
| PaGNN | https://github.com/twitter-research/feature-propagation |
| GCN-LPA | https://github.com/hwwang55/GCN-LPA |
| C&S | https://github.com/CUAI/CorrectAndSmooth |

## A.4 HYPERPARAMETER SETTING

Table 6 shows the hyperparameter we used during our experiments. $\alpha$ is a hyperparameter arising from Label Propagation Zhu (2005) that determines the amount of information absorption from its neighbors. If $\alpha$ is high, it absorbs more information from its neighbors while putting only a small weight, i.e., $1 - \alpha$ to its original label information. For ease of reproducibility, we only searched $\alpha$ in $\{0.8, 0.9, 0.99, 0.999\}$. We fixed temperature, $\tau$ as 0.01 for all datasets and applied a scaled version of PseudoCon loss in two large datasets, Physics and OGBN-Arxiv. As we handled two real-world missing scenarios, uniformly missing and structurally missing, we used different $\lambda$ in some datasets. Here, it is important to note that we did not tune hyperparameters depending on the missing rates on each dataset. As `Goodie` aimed to perform well on both slightly-missing (i.e., abundant features) and severely-missing scenarios (i.e., partial features), we fixed the hyperparameters for each dataset without considering its diverse missing rates.

Table 6: Hyperparameter setting for `Goodie`.

| Dataset | Common | | | Uniform | Structural |
|---|---|---|---|---|---|
| | $\alpha$ | $\tau$ | scaled | $\lambda$ | $\lambda$ |
| Cora | 0.99 | 0.01 | False | 1 | 1 |
| CiteSeer | 0.999 | 0.01 | False | 1 | 1 |
| PubMed | 0.999 | 0.01 | False | 0.0001 | 0.1 |
| WikiCS | 0.9 | 0.01 | False | 10 | 10 |
| Coauthor CS | 0.99 | 0.01 | False | 0.01 | 10 |
| Coauthor Physics | 0.999 | 0.01 | True | 0.00001 | 0.001 |
| OGBN-Arxiv | 0.8 | 0.01 | True | 0.001 | 0.001 |

Table 7: Performance of `Goodie` on link prediction task with other baselines at $mr = 0.9999$.

| Dataset | | Uniformly Missing | | | | Structurally Missing | | | |
|---|---|---|---|---|---|---|---|---|---|
| | | GCN | GCN-NM | FP | Goodie | GCN | GCN-NM | FP | Goodie |
| Cora | AUC | 0.50±0.02 | 0.54±0.04 | 0.82±0.02 | **0.84**±0.02 | 0.50±0.02 | 0.50±0.01 | 0.60±0.12 | **0.84**±0.02 |
| | AP | 0.51±0.01 | 0.55±0.03 | 0.82±0.03 | **0.85**±0.01 | 0.51±0.02 | 0.51±0.01 | 0.59±0.11 | **0.84**±0.02 |
| CiteSeer | AUC | 0.50±0.02 | 0.50±0.04 | 0.75±0.09 | **0.81**±0.03 | 0.48±0.02 | 0.50±0.02 | 0.60±0.12 | **0.81**±0.02 |
| | AP | 0.50±0.01 | 0.53±0.04 | 0.76±0.10 | **0.83**±0.04 | 0.50±0.01 | 0.50±0.02 | 0.59±0.12 | **0.84**±0.02 |
| PubMed | AUC | 0.50±0.02 | 0.59±0.03 | 0.67±0.09 | **0.75**±0.04 | 0.44±0.03 | 0.50±0.03 | 0.64±0.12 | **0.70**±0.04 |
| | AP | 0.52±0.02 | 0.64±0.03 | 0.70±0.10 | **0.79**±0.04 | 0.47±0.02 | 0.50±0.03 | 0.65±0.11 | **0.74**±0.04 |
| Coauthor CS | AUC | 0.68±0.02 | 0.87±0.01 | 0.90±0.02 | **0.93**±0.01 | 0.53±0.01 | 0.54±0.03 | 0.79±0.13 | **0.87**±0.04 |
| | AP | 0.69±0.02 | 0.87±0.02 | 0.88±0.03 | **0.93**±0.01 | 0.53±0.01 | 0.55±0.03 | 0.77±0.12 | **0.86**±0.05 |

## A.5 LINK PREDICTION.

We also conduct an experiment on another downstream task, a link prediction on the severely missing situations, $mr = 0.9999$. Here, we directly compared `Goodie` with GNN-based imputation models on Cora, CiteSeer, PubMed, and Coauthor CS datasets. In Table 7, we have the following observations: **1)** Among zero, neighborhood-mean, and feature propagation imputation methods, feature propagation performed best. This tells us the effectiveness of incorporating neighbors' features when features are severely missed. **2)** Even though FP's performance was close to `Goodie` when features are missed uniformly, FP's performance drop rate became steeper when it comes to structurally missing scenarios. This is because when features are structurally missed, the selected nodes' features will be completely missed which eventually hampers obtaining high-quality embedding used for link prediction. However, `Goodie` could survive and better predict links thanks to its adaptive ability to reflect more of the structure information in such severely missing cases. Moreover, embeddings that contain more of the structure information would further be clustered via additional supervision, i.e., pseudo-labels originated from the LP branch. Here, we notice the importance of incorporating structure information in an adaptive manner is crucial.

## B SENSITIVITY ANALYSIS

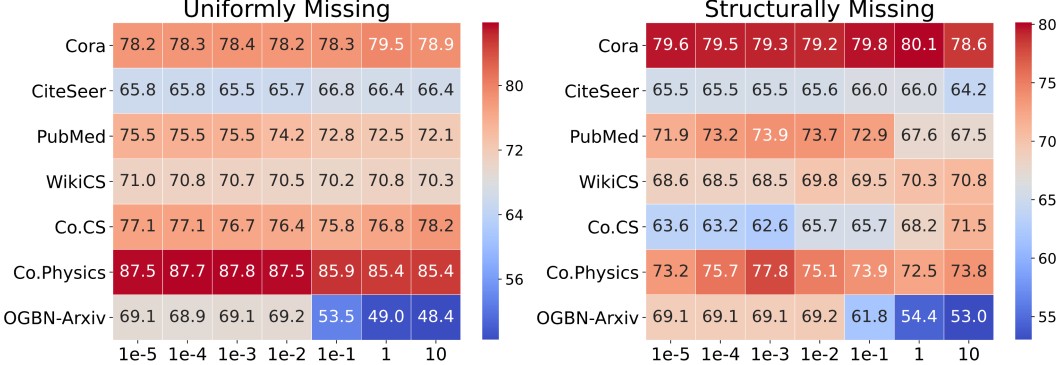

Figure 9: Sensitivity analysis on controlling parameter, $\lambda$. Here the missing rate is set to $mr = 0.9999$, and the matrix element denotes node classification accuracy in each setting.

### B.0.1 SENSITIVITY ON $\lambda$

Figure 9 shows sensitivity analysis on loss controlling parameter, $\lambda$. As `Goodie` consists of two main losses, cross-entropy loss with train labels and pseudo-label contrastive loss, the portion of the latter would grow linearly as $\lambda$ increases. We observe that smaller graph datasets, e.g., Cora, CiteSeer, WikiCS, Coauthor CS have relatively large $\lambda$, whereas larger graph datasets, e,.g., PubMed, Coautrho Physics, OGBN-Arxiv share a small range of $\lambda$. Considering that PseudoCon loss is obtained via node pairs sharing the same label with self-excluded negative pairs, its scale would be proportional to the total number of nodes with its number of pairs, i.e., $\binom{|N|}{2}$. Thus, we conclude that balancing PseudoCon loss that shares range within cross-entropy loss is significant.

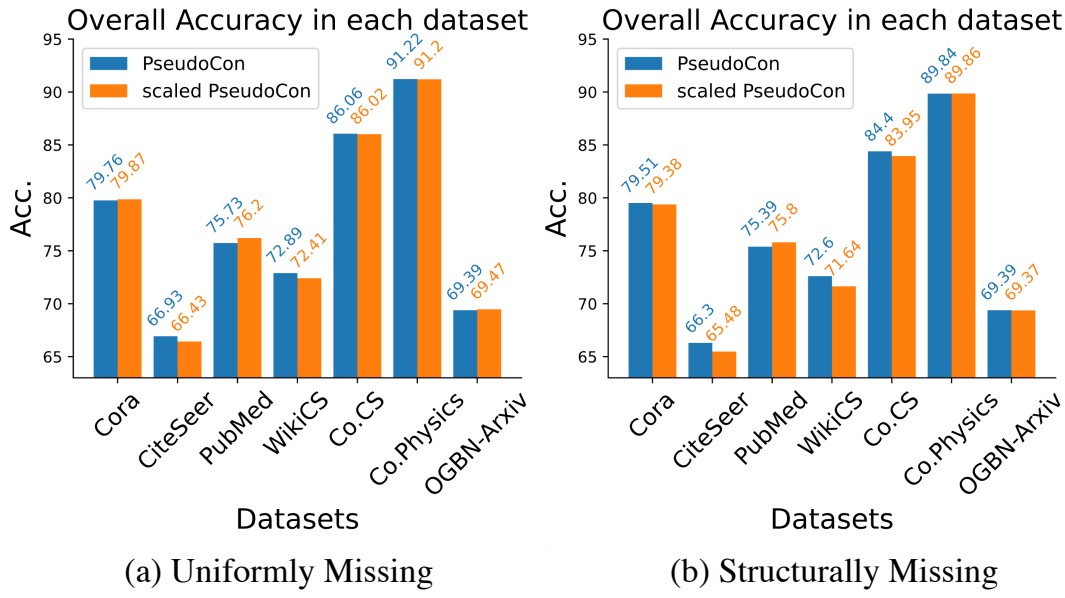

Figure 10: Comparison of PseudoCon loss with its scaled version. To compare with scaled PseudoCon loss, original PseudoCon loss utilized batch sampling technique in Coauthor Physics and OGBN-Arxiv datasets.

### B.0.2 COMPARISON WITH A SCALED VERSION OF PSEUDOCON

As mentioned in Sec 4.3, we additionally proposed a scaled version of PseudoCon loss that effectively reduces the cost, $(\mathcal{O}(N^2) \rightarrow \mathcal{O}(|\mathcal{C}|^2))$. Although it is initially designed to cope with large graph datasets, we further verify its performance on small graph datasets. As shown in Figure 10, we observe that without a huge loss, scaled PseudoCon aligns with small graph datasets as well. In Structurally Missing scenarios, where whole features are deleted for random nodes, we observe original PseudoCon loss generalizes better in relatively small graph datasets, e.g., Cora, CiteSeer, WikiCS, and Coauthor CS. However, when the size of the dataset gets bigger, e.g., PubMed, Co.Physics, and OGBN-Arxiv, original PseudoCon loss was not as effective in small datasets. This is because in such cases, the only feasible way for original PseudoCon loss to cope with large graph datasets is to utilize the batch sampling technique. By doing so, the positive pairs and negative pairs are solely made on that certain batch. Thus, the potential positive and negative pairs, which could be viewed on small graph datasets via full-batch, are not sufficiently utilized. To conclude, when we deal with real-world graphs, it would be preferable to utilize original PseudoCon loss in the small graphs while scaled PseudoCon loss would be preferred in large graphs considering its time and memory complexity.

### B.0.3 EFFECTIVENESS OF PSEUDO-LABEL CONTRASTIVE LEARNING.

To verify our proposed PseudoCon loss (Equation (9)), we compare our loss with based SupCon Khosla et al. (2020) loss with its ablations. Figure 7 shows the effectiveness of PseudoCon with other compared methods. The observations can be summarized as follows: **1) w/o Cont. Loss**: The performance when the contrastive loss is not involved, i.e., $\lambda = 0.0$ (Equation 14), does not belong to the worst case. That being said, carefully incorporating this supervised loss could benefit the training process, which becomes the key challenge. **2) SupCon w/ train**: Naively utilizing SupCon loss with only the given labeled nodes belongs to sub-optimal result. Compared to fully supervised settings, the challenge in semi-supervised settings would be the lack of supervise signals. Thus, solely resorting to the positive pairs made from the given nodes would fall short. **3) PseudoCon (Strong)**: Directly utilizing pseudo-labels originating from the LP branch and regarding its pairs to have equal significance (i.e., 1 as weight) performs worst. More precisely, since pseudo-labels, i.e., the predicted labels, possess their own uncertainty, incorporating such uncertainty would be a necessity. **4) PseudoCon (Weak)**: To further differentiate its positive pairs, we can regard each pair's contribution by giving weights of its multiplied value of each predicted probability (i.e., $\tilde{y}_i * \tilde{y}_p$). This corresponds to the design of Weak Positive as mentioned in Equation (10). We observe this

method outperforms those of **1)**,**2)**, and **3)**. **5) PseudoCon** *(Ours)*: This belongs to our proposed design, which further differentiates Weak Positive by introducing Neutral Positive. Considering there exist three pairs positive pairs, namely, Train-Train (**Strong Positive**), Train-Pseudo (**Neutral Positive**), and Pseudo-Pseudo (**Weak Positive**), we differentiate their degree of contribution as follows: Regarding its probability as an uncertainty measure, we assigned weights as 1 for Strong Positives, $\tilde{y}_\ell$ ($\ell \in \hat{Y}_{pseudo}$) for Neutral Positives, and $\tilde{y}_i * \tilde{y}_p$ for Weak Positives. By doing so, it could achieve the best performance. We also tried the deterministic way, **PseudoCon (Manual)** which is giving fixed weight as 1.0, 0.5, 0.25 in each strong, neutral, and weak positive, but it could not perform well. This again tells us the importance of reflecting uncertainty in each own pair. To further corroborate the effectiveness of PseudoCon, we plot t-SNE Van der Maaten & Hinton (2008) by comparing embeddings obtained without PseudoCon and involving PseudoCon, i.e., `Goodie`. As depicted in Figure 11, we observe embeddings that are obtained via utilizing PseudoCon are better clustered in terms of the same class. Moreover, their decision boundaries became more clear thanks to their negative pairs among differently labeled pairs.

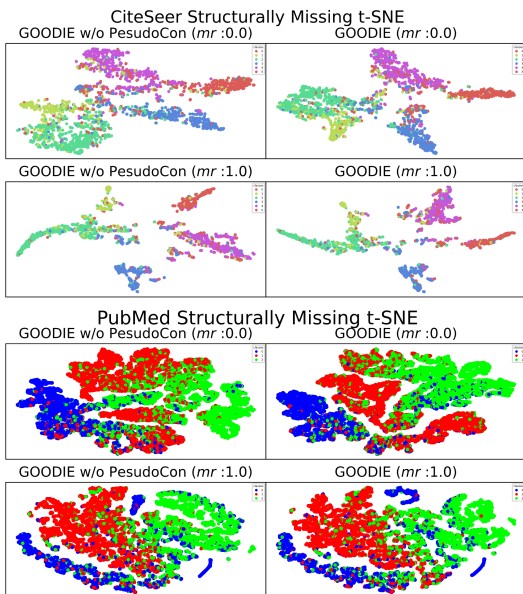

Figure 11: t-SNE plot of CiteSeer and PubMed datasets with their missing rate, 0.0 and 1.0. After utilizing PseudoCon loss, nodes in the same class become more clustered while nodes in different classes become distinctive.

