# OpenReview forum: "Oldie but Goodie: Re-illuminating Label Propagation on Graphs with Partially Observed Features"
_ICLR.cc/2025/Conference — ICLR 2025 Conference Withdrawn Submission_

### Official Review · Reviewer_NGZ1 · 2024-10-28

**Soundness:** 2
**Presentation:** 3
**Contribution:** 1
**Rating:** 3
**Confidence:** 4

**Summary:**

This paper focuses on combining the two classical graph methods, Label propogation and Feature propagation, for partial feature/label graph learning. The proposed Goodie model provides a trainable framework that benifits from neighborhood structure and feature information with a gated fusion of LP and FP outputs, as well as a Pseudo-Label Contrastive Learning loss. Experimental results validate the performance increments of the proposed Goodie framework.

**Strengths:**

+ The method combines two fundamental parameter-free graph method and achieves incremental performance.
+ The proposed model is well presented and easy to follow.
+ Detailed studies on the model performance w.r.t. different feature missing settings.

**Weaknesses:**

- The proposed Goodie appears to be a rather straightforward fusion of existing label and feature propagation models.
- The introduced contrastive learning loss provides limited improvement according to the ablation study in Appendix B.
- The performance of Goodie is merely compared with several basic baseline methods that were proposed before 2021. It seems that the performance of Goodie is grounded by the LP and FP model, yet fails to make any significant improvement over on them.
- Several typos and non-standard reference format. The citations in the paper are not correctly enclosed by brackets.

**Questions:**

Please refer to the weaknesses.

---

### Official Review · Reviewer_GCzP · 2024-10-29

**Soundness:** 2
**Presentation:** 2
**Contribution:** 2
**Rating:** 3
**Confidence:** 5

**Summary:**

This paper investigates graph representation learning under missing features, since sensitive information could be missing due to privacy concerns. The authors claim that existing models perform worse than traditional structure-based models when the missing rate is high. To address this issue, the authors propose to combine the classical label propagation with feature propagation. Specifically, the output of predictions of LP are mapped to hidden embeddings that align with FP. Some pseudo labelling and contrastive learning strategies are utilized to optimize the model’s parameters. Experimental results on 7 public datasets demonstrate that the proposed model could achieve good performance compared with existing baselines.

**Strengths:**

1.	The paper is generally well-written and easy to understand.

2.	Both large-scale and small datasets are utilized to evaluate the performance of the proposed model.

3.	Ablation studies are conducted to show the effectiveness of the proposed components.

**Weaknesses:**

1.	The novelty of the proposed model is only incremental, as it simply combines label propagation and feature propagation with attention mechanism. There are no new modules in the LP and FP branches. The contrastive learning module is also from SupCon.

2.	The experiments are only conducted on homophily graphs, and it is not clear how it will perform on heterophily graphs.

3.	The performance improvement is only marginal, and the authors did not explain why their model does not work on Coauthor datasets.

**Questions:**

1.	The novelty of the proposed model is only incremental, as it simply combines label propagation and feature propagation with attention mechanism. There are no new modules in the LP and FP branches. The contrastive learning module is also from SupCon.

2.	The experiments are only conducted on homophily graphs, and it is not clear how it will perform on heterophily graphs.

3.	The performance improvement is only marginal, and the authors did not explain why their model does not work on Coauthor datasets.

4.	The used baselines are too old, and some key baselines are missing. Please kindly refer to the following references [1] [2] [3].

5.	According to Figure 5 and Figure 7, most of the proposed modules do not contribute too much when the observed rates are high.

6.	Why does $\tilde{y}$ need to divide $\tau$ in Eq (10)? Meanwhile, it is not necessary to constrain the value of $w_{ip}$ into the range of [0,1].

References:

[1] Jiang X, Qin Z, Xu J, et al. Incomplete graph learning via attribute-structure decoupled variational auto-encoder[C]//Proceedings of the 17th ACM International Conference on Web Search and Data Mining. 2024: 304-312.

[2] Huo C, Jin D, Li Y, et al. T2-gnn: Graph neural networks for graphs with incomplete features and structure via teacher-student distillation[C]//Proceedings of the AAAI Conference on Artificial Intelligence. 2023, 37(4): 4339-4346.

[3] Tu W, Xiao B, Liu X, et al. Revisiting initializing then refining: an incomplete and missing graph imputation network[J]. IEEE Transactions on Neural Networks and Learning Systems, 2024.

**Details Of Ethics Concerns:**

NA.

---

### Official Review · Reviewer_chfv · 2024-11-02

**Soundness:** 2
**Presentation:** 2
**Contribution:** 2
**Rating:** 5
**Confidence:** 3

**Summary:**

This paper investigates the graph learning problem with GNNs under the situation of missing features. A method termed Goodie is proposed, which unified the label propagation and feature propagation, along with a contrastive learning objective with a pseudo label mechanism. Experiments are conducted to show the effectiveness of the proposed method.

**Strengths:**

1. Some specific designs in the proposed method is interesting. Specifically: 1. The structure-feature attention is useful in combining the results of feature and label propagation; 2. The fine-grained consideration of pseudo label is practical.
2. The idea of balancing the contribution of structure and feature information makes sense, especially when the missing situation is unknown or uncertain.

**Weaknesses:**

1. The idea of combining feature propagation and label propagation is not innovative enough. Previous papers [*1,*2] also consider similar idea.
[*1] Wang, Yangkun, et al. "Why propagate alone? parallel use of labels and features on graphs." arXiv preprint arXiv:2110.07190 (2021).
[*2] Shi, Yunsheng, et al. "Masked label prediction: Unified message passing model for semi-supervised classification." arXiv preprint arXiv:2009.03509 (2020).
2. Key baselines lack. There are some GNNs that are also designed for the situation of missing data, such as [*3,*4]. However, in the experiment part, the authors only compared the proposed method with some out-of-date methods. More up-to date baselines are expected.
[*3] Huo, Cuiying, et al. "T2-gnn: Graph neural networks for graphs with incomplete features and structure via teacher-student distillation." Proceedings of the AAAI Conference on Artificial Intelligence. Vol. 37. No. 4. 2023.
[*4] Liu, Yixin, et al. "Learning strong graph neural networks with weak information." Proceedings of the 29th ACM SIGKDD Conference on Knowledge Discovery and Data Mining. 2023.
3. A basic assumption of the proposed method seem like "homophily assumption": connected nodes have similar feature & labels. In this case, will the proposed method also work well on heterophily graphs? If yes, please attach more results on heterophilic datasets; if not, it's better to point out and discuss this limitation.

**Questions:**

1. The proposed method seems to be sensitive to the quality and quantity of labels and pseudo labels. In this case, how will it perform when the labels are scarce or noisy?
2. The SupCon itself can also generate classification results directly. Then should we still require the GNN-based classifier?

---

### Official Review · Reviewer_dunz · 2024-11-03

**Soundness:** 2
**Presentation:** 2
**Contribution:** 1
**Rating:** 3
**Confidence:** 4

**Summary:**

This paper introduce a framework called Goodie that predict unlabeled nodes on graphs with partial features. Goodie incorporates label propagation (LP) and feature propagation (FP). Goodie leverages structure-feature attention to control the weights of the LP branch and the FP branch. Goodie further utilize pseudo-label contrastive loss to fully use the potential of the LP branch.

**Strengths:**

* Significance: Addressing the missing data problem is important.

* Presentation: Overall, this paper is well-written.

**Weaknesses:**

* Misrepresentation of Information: Stating that the outputs of the LP branch and the FP branch represent structural information and feature information is inaccurate. The LP branch utilizes class/label information, while the FP branch leverages feature information, and both branches rely on structural information through the propagation process.

* Only Node Classification: Goodie is limited to addressing only node classification, which restricts its applications.

* Marginal Performance Gains: The performance gains by Goodie seems very marginal in most observed rates, only except extremely high missing rates.

* Missing Amazon Datasets: Why did the authors not compare the methods on the Amazon datasets used in FP's evaluation?

* Artificial Settings: The observed scenarios at 0.0001 (0.01%) seem to be highly artificial settings, not reflective of the real world.

* Lack of State-of-the-Art Comparison: FP is not the state-of-the-art method. Comparisons with PCFI, a recent study following FP, have been omitted. SAT, an important baseline, is also not included in the comparisons.

**Questions:**

See weaknesses.

---

### Note · Authors · 2024-12-02

I have read and agree with the venue's withdrawal policy on behalf of myself and my co-authors.